# An Analysis of the Impact of Religious Affiliation and Strength of Religiosity on Sexual Health Practices of Sexually Active Female College Students

**DOI:** 10.3390/ijerph20227075

**Published:** 2023-11-17

**Authors:** Emily Glazer, Emma Valdez, Justin A. DeBlauw, Stephen J. Ives

**Affiliations:** Health and Human Physiological Sciences, Skidmore College, Saratoga Springs, NY 12866, USAjdeblauw@skidmore.edu (J.A.D.)

**Keywords:** religion, contraceptives, women’s health, reproductive health

## Abstract

Despite great strides in the development of contraceptive technologies, the United States has one of the highest teen pregnancy rates in the world. Religion and associated values may shape the sexual health behaviors of college students, as prior studies have aimed to determine how social factors may influence the use of contraception amongst college students. Thus, we sought to examine the differences in current contraceptive methods and the age of first contraceptive usage among sexually active female college students with different religious affiliations and strengths of religiosity. It was hypothesized that there would be no difference in current contraceptive methods among different religious affiliations and strengths of religions and that there would be a difference in the age of first contraceptive usage among different religious affiliations and strengths of religiosity. Two hundred and twenty-four college-aged females completed a 20-question survey about sexual health and religious practices. Chi-squared tests were implemented to determine the frequencies of responses across religious affiliations and strengths of religiosity. Significant differences in the frequency of responses for the age of first contraceptive usage were observed across different strengths of religiosity (*p* = 0.016) and for the self-perceived impact of religion on sexual health across different religious affiliations (*p* = 0.033) and strengths of religiosity (*p* = 0.005). All other differences were found not to be statistically significant. It was determined that increased strengths of religiosity resulted in delayed onset of contraceptive usage and that both different religious affiliations and greater strengths of religiosity lead to different self-perceived impacts of religion on sexual health despite low levels of current practice.

## 1. Introduction

Religion has been a central part of the way individuals live their lives since humans became civilized. Today, across the world, 31% of people practice Christianity, 24% practice Islam, and 15% practice Hinduism [1]. Religion is a central part of our society and its pillars help shape the way we live our lives. In the United States, church and state are separate, yet the choices of citizens, such as family planning, are often shaped by religion. Practices such as abstinence or being against abortion are rooted in religion. Yet, world wide in 2019, among the 1.9 billion Women of Reproductive Age group (15–49 years), only 1.1 billion reported having a need for family planning, and of these, only 842 million are using contraceptive methods, and 270 million felt as if they had an unmet need for contraception [2]. While contraception may be viewed as a sin in some religions, birth control reduces pregnancy-related morbidity and mortality, risk of certain cancers, and severe menstruation symptoms. On a large scale, contraception can also improve a country’s economy unemployment levels, and reduce the population. College campuses, like Skidmore College, often have programs to promote sexual health and provide education and free samples of condoms, lubrication, and other sexual health products. This is valuable due to the high rates of sexual activity on college campuses. The 2018 National College Health Assessment found that 66% of students had sex in the last 12 months [3]. But what makes those sexual health programs effective? College is a time in which young adults are often making decisions for themselves for the first time, contraception usage included. It is important to understand what is affecting the student body’s decisions regarding safe sex. Most religions enforce moral behavior via heaven/hell, karma, or scriptures, and critically, contraception is often viewed, either overtly or indirectly, as a sin or moral misstep, where abstinence, particularly until marriage, may be the only morally acceptable choice with specific regards to sexual health.

Additionally, many previous studies have shown that the simple act of attending religious worship, independent of religious affiliation, has resulted in increased measures of well-being [4,5]. More specifically, these past studies have shown that increased religious commitment has been directly related to decreased stress, substance use, teen pregnancy, crime, welfare dependency, and suicide, as well as increased happiness, self-esteem, marital satisfaction, and physical health [4,5]. Therefore, it is possible that, based on the relationship between religiosity and teen pregnancy rates, religion, or religious upbringing in the absence of current practice, may have an important and measurable impact on the overall sexual health of college students. 

Therefore, the purpose of this study was to examine the differences in contraceptive methods and the age of first contraceptive usage among sexually active female students with different religious affiliations and strengths of religiosity at a liberal arts college. It was hypothesized that there would be no difference in the contraceptive method of choice among sexually active female students with different religious affiliations and different strengths of religions. In addition, it was hypothesized that there would be a difference in the age of first contraceptive usage among sexually active female students with different religious affiliations and different strengths of religiosity.

## 2. Materials and Methods

### 2.1. Subjects and General Procedures

Participants were college-aged (18–22 years old) female students enrolled full-time at Skidmore College in Saratoga Springs, New York. Since the present study follows a survey methodology, all eligibility was determined post-survey completion. Participants were excluded if they were not within the age range of 18–22 years, were not enrolled full-time as a student, self-reported their biological sex as male, or if they self-reported to be non-sexually active (sexual activity defined as engaging in oral sex or vaginal/anal penetration at least 3 times in the last 3 months). This definition of sexual activity is based on the Centers for Disease Control (CDC) sexual risk behaviors, which can result in unintended pregnancy or sexually transmitted disease. Based on the approximate 1534 self-identified female students at Skidmore College, a 95% confidence level, and a 5% margin of error, the goal was to enroll 307 female Skidmore College students. After applying the exclusion criteria, there were 224 final participants who completed this study (Figure below). Participants provided written informed consent prior to participating in the survey. This study was reviewed and approved by the Skidmore College Institutional Review Board (IRB# 2201-1013). 

### 2.2. Procedure

Participants were recruited via email, poster advertising, and word-of-mouth to complete the survey. The Skidmore Health Promotion office sent the link to the survey using an all-class list to ensure the survey was accessible to everyone. There were 20 questions in total, both multiple-choice and fill-in-the-blank. Fill-in-the-blank options were included in order to provide a chance for any response to be included that was not a multiple-choice option. The survey inquired about several topics, including inclusion criteria, demographic information, personal religious information, and personal sexual health information. Inclusion criteria questions allowed for the participants to answer about their status as a Skidmore student, age, biological sex determined at birth, and sexual activity. To be included, participants must have been assigned female biological sex at birth. Participants were considered sexually active if they had penetrative sex (i.e., oral, vaginal, etc.) 3 times in the last 3 months. Demographic variables included class year, race, self-identified gender identity, and nationality. Religious questions prompted the participants to answer what religion they identified with and how often they practiced that religion. The strength of religiosity for each participant was graded on a scale of how often they practiced that religion. Finally, sexual health questions inquired about the student’s age, first sexual activity, current contraceptive method, age of first contraceptive use, comfort discussing sexual health with a parent, comfort discussing sexual health with a healthcare provider, and personal belief of religion on sexual health practices. The survey was distributed over a 2-week period in the spring of 2022. 

### 2.3. Statistical Analysis

Statistical analysis was carried out using online open-source software (JASP, v 0.13.1, Amsterdam, The Netherlands). A Chi-Square analysis was used to determine the probability of differences between the religious variables and reproductive health variables. The level of significance was set at *p* < 0.05. Data are presented as descriptive statistics, or means and standard deviations, as appropriate.

## 3. Results

### 3.1. Subject Characteristics

As seen in Figure 1, the distribution of graduation year for all survey respondents was relatively even, with the majority of respondents (31%) graduating in the year 2022 (Figure 2). In addition, the majority of respondents self-identified as white (81%), cisgender females (88%) who were born in the United States of America (93%; Figure 3, Figure 4 and Figure 5). In regard to the distribution of religious affiliations and strengths of religiosity, Figure 6 demonstrates that the majority of respondents identified as having no religious affiliation (44%), followed by being Jewish (20%) or Catholic (16%) (Figure 6). All other religious affiliations recorded a distribution of less than 10% (Figure 6). Finally, a downward trend in the frequency of religious practice was observed in Figure 7. The majority of respondents never practice their religion (47%), followed by practicing once or twice a year (28%), at least five times a year (13%), and then once a month, week, or day with distributions of less than 10% (Figure 7). 

### 3.2. Difference in Religious Affiliation and Sexual Health Variables

As noted in Table 1 and Table 2, there were no significant differences in the frequencies of responses for both the current contraceptive method used and the age of first sexual intercourse across different religious affiliations (contraceptive method: Chi^2^ (56, 224) = 33.1, *p* = 0.994; age first sex: Chi^2^ (16, 224) = 18.2, *p* = 0.311; Table 1 and Table 2). Similarly, there were no significant differences in the frequencies of response for the age of first contraceptive use across different religious affiliations (Chi^2^ (24, 224) = 22.6, *p* = 0.546; Table 3). There were also no significant differences observed in the proportion of responses for comfort discussing sexual health with a parent or with a healthcare provider across different religious affiliations (parent: Chi^2^ (32, 224) = 24, *p* = 0.843; healthcare provider: Chi^2^ (32, 224) = 40.2, *p* = 0.152; Table 4 and Table 5). Finally, however, there were significant differences in the proportion of responses for the self-perceived impact of religion on personal sexual health practices across different religious affiliations (Chi^2^ (32, 224) = 48.2, *p* = 0.033; Table 6).

### 3.3. Difference in Religiosity and Sexual Health Variables

In regard to the impact of religiosity on sexual health practices, Table 7 shows that there were no significant differences in the frequencies of responses for both current contraceptive method used and the age of first sexual intercourse across different strengths of religiosity (contraceptive method: Chi^2^ (35, 224) = 37.9, *p* = 0.339; age first sex: Chi^2^ (10, 224) = 13.03, *p* = 0.222; Table 7 and Table 8). There was, however, a significant difference in the proportion of responses for the age of first contraceptive use observed across different strengths of religiosity (Chi^2^ (15, 224) = 29.1, *p* = 0.016; Table 9). In addition, there were no significant differences in the proportion of responses for both comfort discussing sexual health with a parent or with a health care provider across different strengths of religiosity (parent: Chi^2^ (20, 224) = 17.3, *p* = 0.633; healthcare provider: Chi^2^ (20, 224) = 12.03, *p* = 0.915; Table 10 and Table 11). Finally, there was a significant difference in the frequency of responses for the self-perceived impact of religion on personal sexual health practices observed across different strengths of religiosity (Chi^2^ 20, 224) = 40.06, *p* = 0.005; Table 12).

## 4. Discussion

The present study sought to determine the differences in the current contraceptive method of choice and the age of first contraceptive use among sexually active female students at a liberal arts college with different religious affiliations and strengths of religiosity. It was hypothesized that no differences would be observed in the current contraceptive method of choice among the sexually active female students with both different religious affiliations and strengths of religiosity. It was also hypothesized that there would be a significant difference in the age of first contraceptive use across both different religious affiliations and strengths of religiosity.

The majority of the survey respondents were white, cisgender female students at Skidmore College who had no religious affiliations and had low strengths of religiosities, measured by never practicing their religion in the last calendar year. It was determined that there were no differences in the frequencies of responses for the current contraceptive method, age of first sexual intercourse, age of first contraceptive use, and comfort discussing sexual health with either a parent or health care provider across different religious affiliations. It was also observed that there were no differences in the frequencies of responses for the current contraceptive method, age of first sexual intercourse, and comfort discussing sexual health with either a parent or health care provider across different strengths of religiosity. Despite these findings, it was determined that there were significant differences found in the age of first contraceptive usage across different strengths of religiosity, self-perceived impact of religion on personal sexual health practices across different religious affiliations, and self-perceived impact of religion on personal sexual health practices across different strengths of religiosity. Thus, religious upbringing prior to college may play a role in the onset of sexual health practices and decisions in those practices.

### 4.1. Participant Demographics

It was observed in the present study that the majority of the participants self-identified as white, cis-gendered females who were born in the United States (Figure 3, Figure 4 and Figure 5). These demographics were expected, as Skidmore College, a private undergraduate institution located in Saratoga Springs, NY, is a historically all-women’s liberal arts college and is classified as a predominantly white institution (PWI). As noted by the Admissions Office at Skidmore College, the percentage of female students for the class of 2024 is 59%, while the percentage of males is 41% [6]. Throughout the United States, the majority of liberal arts institutions are seen to have a gender ratio of 60:40 female to male students, as many of these schools, like Skidmore, were founded in the mid-20th century as all women’s schools to provide equal access to higher education to all and thus continue to have a higher percentage of female students compared to male. In addition, as a PWI, the Admissions Office noted that in the class of 2024, only 26% of students identified as people of color, breaking down as follows: 9.6% Latinx, 5.4% Asian, 5.4% two or more races, and 5% black [6]. Therefore, both the racial and gender demographics observed in the present study were expected based on the known demographics of Skidmore College.

It was also observed that the majority of participants self-identified as having no religious affiliations and having never practiced religion in the last calendar year, thus resulting in having a low strength of religiosity (Figure 6 and Figure 7). At Skidmore College, limited data is available regarding the religious demographics of the student body. However, because religiously affiliated clubs at Skidmore College tend to have over 100 members and are one of the most popular forms of extra-curricular activities for Skidmore students, it was assumed by the researchers that the majority of students would be affiliated with either a Christian affiliation or Judaism, rather than identify as unaffiliated because of the popularity of these campus organizations. However, based on previous research examining the ways in which age impacted religious affiliation, it was noted in *The 2020 Census for American Religion*, published by the Public Religion Research Institute (PRRI), that the American adolescent population (ages 18–29) have the greatest counts of ‘unaffiliated’ responses out of all respondent age groups and these counts are only increasing as time progresses [7]. Specifically, in 1986, the PRRI reported that only 10% of 18–29-year-old respondents identified as unaffiliated, but in the years 1996, 2006, 2016, and 2020, the percentages rose to 20%, 23% 38%, and 36%, respectively [7]. Therefore, while the present study recorded percentages of religious affiliation higher than what was found by PRRI in 2020, the general trend that college-aged students are likely to be religiously unaffiliated adheres to previous research. 

### 4.2. Impact of Religious Affiliation

As previously described, the only significant difference observed in the frequencies of responses across different religious affiliations was in regard to the self-perceived impact of religion on their own sexual health practices (Table 6); this suggests that female college students of different religious affiliations perceived that their respective religions varied in importance as a social factor that shapes sexual health practices. Interestingly, however, no differences were found across the religious affiliations and the proportion of responses for the age of first sex, current contraceptive method, age of first contraceptive use, and the comfort of discussing sexual health with either a parent or a healthcare provider. Thus, based on the data, the participants’ religious affiliations were not observed to impact the sexual health variables of interest in the present study, even though the participants believed that they had. No previous studies have examined the perceived impacts of religion on sexual health related to the statically determined impact of religion. Therefore, not only is this conclusion novel, but it also introduces a new area of study where further research is warranted, especially considering that current religious practice, or not, seems to play a role in sexual health decisions.

The research that has been conducted previously regarding the impact of specific religious affiliations on the sexual health practices of adolescents has led to many contrasting results. In general, these studies solely focused on how different Christian religions (Catholicism and Protestantism) and non-Christian religions (Judaism and Islam) influenced sexual activity in general. In one study, Gold et al. determined that adolescents who identified with non-Christian religions were the most sexually active, with only 4.8% responding that they had never had sex, vs. 13% for Catholics, 27.2% for Baptists, 28.9% for other Christians, and 23.9% for those with no affiliation [8]. In contrast, the study by Manlove et al. determined that Jewish and other non-Christian adolescents were the least sexually active compared to Christian respondents, with only 41% of Jewish and 44% of teens self-designating without religious affiliation responding that they had sex before age 18 [2]. While the present study only adds to the contradictory results, as no differences were found in those identifying amongst the different affiliations, the previous research on this topic is extremely broad and limited; thus, further research is warranted in order to better understand these influences.

### 4.3. Impact of Strength of Religiosity

Unlike the impact of religious affiliation, which was only found to impact one sexual health variable significantly, it was observed that the impact of different strengths of religiosity, independent of affiliation, resulted in significantly different proportions of responses for both ages of first contraceptive use and the self-perceived impact of religion on sexual health practices. Specifically, it was observed that an increased strength of religiosity, measured by a greater frequency of religious practice, resulted in delayed onset of contraceptive usage (Table 9). However, no differences in the frequencies of responses for age of first sex were observed across different strengths of religiosities, thus demonstrating that religiosity had no impact on the sexual activity of female college students (Table 8). This conclusion is contested by previous research, as Cotton and Berry 2007 determined that adolescents who attended religious services of any denomination frequently, as defined by the attendance of greater than once per week, were less sexually active than those who did not attend as often, as only 39% of adolescents who attended religious services frequently reported having sexual intercourse in the last year compared to 65% of respondents who did not frequently attend services [5]. It is possible that in the transition from adolescence to college, these relations may be modified and could explain the difference between the prior literature and current findings. 

Therefore, it can be concluded that religiosity impacts the age of first use of contraception. However, not the age of first sexual activity in college females, religiosity, independent of affiliation, does not prevent sexual activity in itself, but it does limit the use of contraception during first sex, thus limiting safe sex practices. This conclusion has been found in previous research. Based on the results of the US Religion Landscape survey conducted by the CDC in 2008, it was determined that teens of the 21st century are extremely sexually active and thus participate in sexual behaviors regardless of their religiosity, as a positive high correlation (r = 0.73) was found between increased young adult religiosity and teen birth rates [9]. Therefore, this study demonstrated that highly religious teens engage in sexual activities to a similar degree as their non-religious peers yet were highly influenced by their religion to not use birth control while engaging in these similar behaviors, thus resulting in higher rates of teen pregnancy among the highly religious participants [9]. The historical written scriptures and/or their interpretation highlight sexual practice outside of marriage as turpitude, thus likely explaining the predilection to avoiding contraceptives in those who are religious. 

Similarly, to the conclusion found between the differences in religious affiliations and the self-perceived impact of religion on sexual health practices, it was concluded in the present study that increased strengths of religiosity, measured by a greater frequency of religious practice, led to different self-perceived impacts of religion on the sexual health practices of college female students (Table 12). This observation suggests that independent of religious affiliation, as the strength of religiosity increases, the respondents believed that their respective religions were an increasingly important social factor that influenced their personal sexual health practices. Unlike the previous conclusion regarding religious affiliations in which no other differences were found to be significant, and thus there was a contradiction between perceived impact and actual impact, the frequencies of responses for age of first contraceptive use were significantly different across different strengths of religiosity. This observation novel reveals that, as previously stated, religiosity impacts when female college students start using birth control; it also reveals that the self-perception of this impact stands true to tested measures. Once again, however, because no previous literature was found to also examine how perceived impacts of religion on sexual health related to the statically determined impact of religion, future research on this topic is warranted.

### 4.4. Limitations and Future Research

The present study was not completed without limitations. First, because many responses were excluded from data analysis, the present study had a smaller sample size than anticipated. Due to human error while completing the survey, many respondents did not answer all the questions or answered all questions but forgot to hit the ‘submit’ button; thus, their responses were omitted from the analysis. It was more common, however, for completed survey responses to be excluded because the participants did not match the inclusion criteria. Many participants who were either of the male sex, assigned at birth, or were not sexually active based on the study’s definition submitted responses that could not be used, as the current study focused on those who were assigned the female sex at birth and were recently sexually active. We defined sexual activity based on the CDC sexual risk behaviors, given the potential for unintended pregnancy or sexually transmitted disease. Another limitation of this study was the religious demographic of the sample size. Skidmore College is not a religiously affiliated institution. In addition, the majority of the participants had no religious affiliations and never practiced a religion in the last calendar year. Therefore, by pooling at both a liberal institution that has no ties to religious practices and within a study body that is not religious, it was difficult to study the impact of a certain identity on sexual health practices if the majority of respondents did not have the identity in question.

Based on these limitations and the limited previous research available on the religion and sexual health of college students, future research is warranted. Particularly, repeating the present study’s methodology at a religiously affiliated institution, such as the Catholic University of America or Yeshiva University, would ensure that the participant pool is more religious, and including those who are abstinent could provide additional critical insight into the relation between religious practice and knowledge and use of contraceptives. In addition, it would be beneficial to repeat this study’s methodology at another institution that is more diverse, regardless of whether it is religiously affiliated. As a PWI, the participants of the present study from Skidmore College were majority white, cis-gendered females. It is impossible to generalize these results to all female college students, as the experiences of these respondents may differ from those of different racial and gender identities. Therefore, repeating this study at a more diverse institution, such as a public university or one located in a different geographic area, a more diverse participant pool may be used and thus make the conclusions generalizable to more identities.

## 5. Conclusions

We sought to determine how religious affiliation and strength of religiosity impacted the sexual health behaviors of college-aged sexually active females. It was determined that an increase in the strength of religiosity was associated with the age of first contraceptive usage but not the age of first sex, thus demonstrating that religiosity does not delay adolescent sexual activity but does delay the use of contraception during first sex. Further, it was observed that both different religious affiliations and greater strengths of religiosity lead to different self-perceived impacts of religion on sexual health behaviors, demonstrating that college-ages females perceive religion to be an important social factor that shapes their sexual health behaviors, even in those who may not be currently practicing their religion. 

## Figures and Tables

**Figure 1 ijerph-20-07075-f001:**
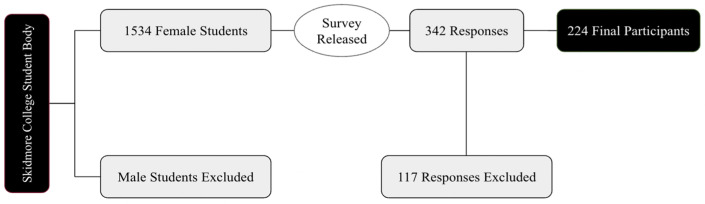
Overview diagram of data collection and analysis.

**Figure 2 ijerph-20-07075-f002:**
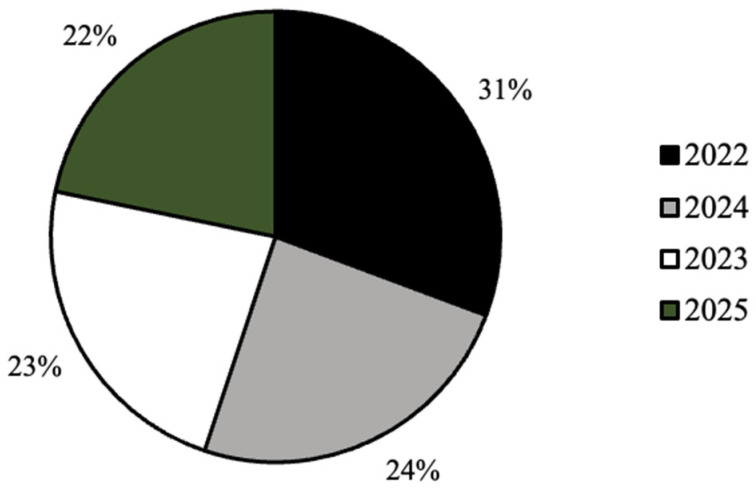
Distribution (%) of graduation year for all survey respondents (*n* = 224).

**Figure 3 ijerph-20-07075-f003:**
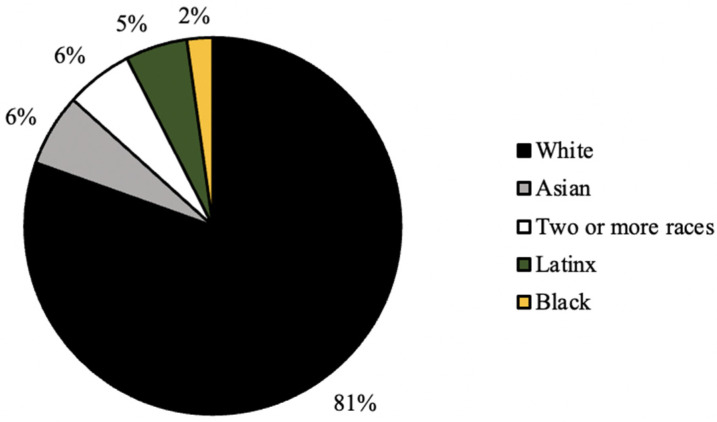
Distribution (%) of racial identity for all survey respondents (*n* = 224).

**Figure 4 ijerph-20-07075-f004:**
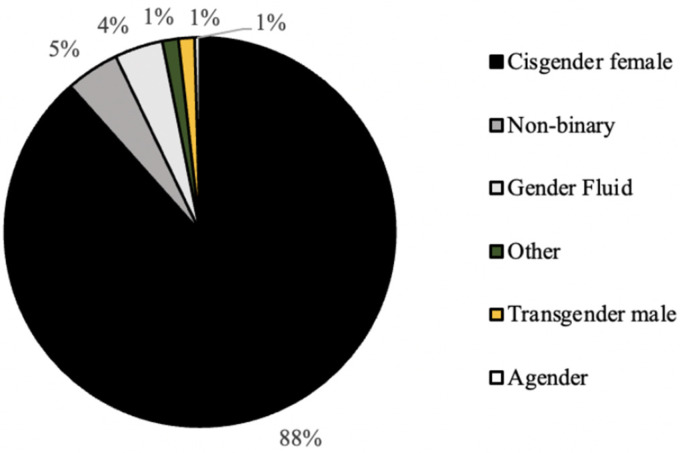
Distribution (%) of self-identified gender identity for all survey respondents (*n* = 224).

**Figure 5 ijerph-20-07075-f005:**
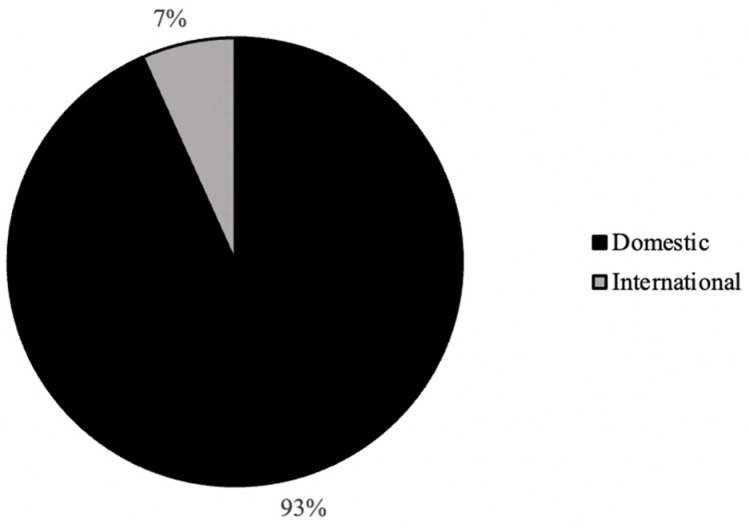
Distribution (%) of nationality status for all survey respondents (*n* = 224).

**Figure 6 ijerph-20-07075-f006:**
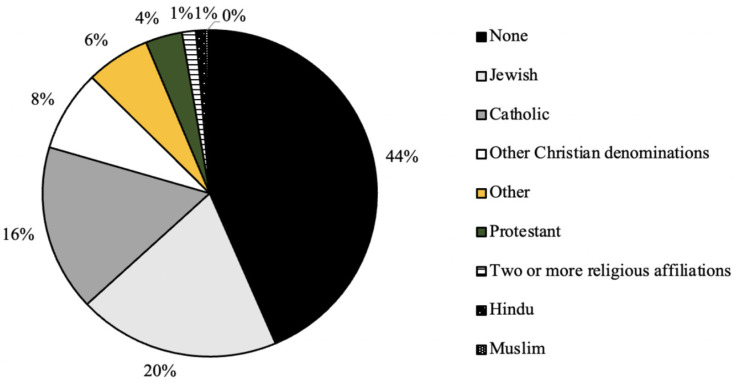
Distribution (%) of religious affiliations for all survey respondents (*n* = 224).

**Figure 7 ijerph-20-07075-f007:**
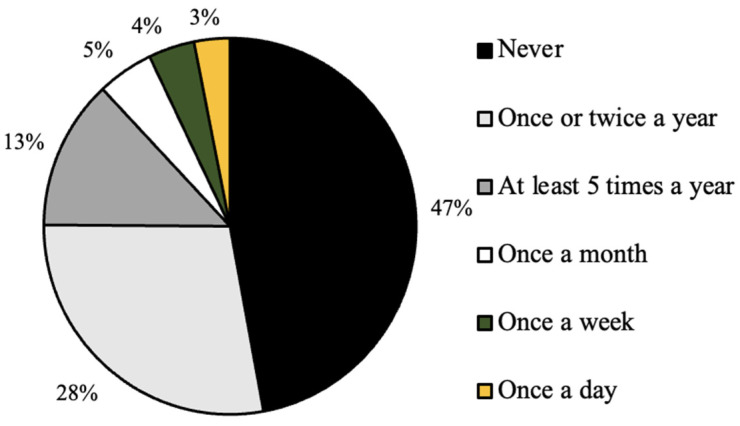
Distribution (%) frequency of religious practice for all survey respondents (*n* = 224).

**Table 1 ijerph-20-07075-t001:** Religious affiliation and current contraception method contingency table.

Religious Affiliation	Current Form of Contraception Method								
	None	Condoms	Pill	IUD	Implant	Plan B	Multiple forms	Other	Total
None	7	13	16	17	7	0	37	1	98
Catholic	1	6	6	10	0	0	14	0	37
Protestant	1	2	1	0	1	0	3	0	8
Other Christian	1	1	1	4	1	0	9	0	17
Jewish	5	4	7	4	2	1	20	1	44
Muslim	0	0	0	1	0	0	2	0	3
Hindu	0	0	0	0	0	0	2	0	2
Two or More	1	0	0	1	0	0	1	0	3
Other	1	1	1	2	0	0	7	0	12
Total	17	27	32	39	11	1	95	2	224

Chi^2^ (56, 224) = 33.1, *p* = 0.994.

**Table 2 ijerph-20-07075-t002:** Religious affiliation and age of first sex contingency table.

Religious Affiliation	Age of First Sex			
	Prior to 15	15 to 18	18 to 22	Total
None	7	65	26	98
Catholic	3	21	13	37
Protestant	0	3	5	8
Other Christian	2	12	3	17
Jewish	5	26	13	44
Muslim	0	1	2	3
Hindu	0	2	0	2
Two or More	1	1	1	3
Other	3	4	5	12
Total	21	135	68	224

Chi^2^ (16, 224) = 18.2, *p* = 0.311.

**Table 3 ijerph-20-07075-t003:** Religious affiliation and age of first contraception use contingency table.

Religious Affiliation	Age of First Contraception Usage				
	Never	Prior to 15	15 to 18	18 to 22	Total
None	6	6	63	23	98
Catholic	1	1	21	14	37
Protestant	1	0	2	5	8
Other Christian	1	1	9	6	17
Jewish	2	5	22	15	44
Muslim	0	0	1	2	3
Hindu	0	0	1	1	2
Two or More	1	0	1	1	3
Other	0	2	5	5	12
Total	12	15	125	72	224

Chi^2^ (24, 224) = 22.6, *p* = 0.546.

**Table 4 ijerph-20-07075-t004:** Religious affiliation and comfort discussing sexual health with parent/guardian contingency table.

Religious Affiliation	Comfort Discussing Sexual Health with Parent or Guardian					
	Extreme Discomfort	Mild Discomfort	Neither Discomfort nor Comfort	Mild Comfort	Extreme Comfort	Total
None	16	33	10	25	14	98
Catholic	9	15	3	7	3	37
Protestant	3	1	1	2	1	8
Other Christian	2	5	1	7	2	17
Jewish	5	14	6	12	7	44
Muslim	2	0	0	1	0	3
Hindu	0	2	0	0	0	2
Two or More	1	1	0	0	1	3
Other	2	4	0	4	2	12
Total	40	75	21	58	30	224

Chi^2^ (32, 224) = 24, *p* = 0.843.

**Table 5 ijerph-20-07075-t005:** Religious affiliation and comfort discussing sexual health with healthcare provider contingency table.

Religious Affiliation	Comfort Discussing Sexual Health with Healthcare Provider					
	Extreme Discomfort	Mild Discomfort	Neither Discomfort nor Comfort	Mild Comfort	Extreme Comfort	Total
None	3	4	11	48	32	98
Catholic	1	5	4	17	10	37
Protestant	0	0	3	4	1	8
Other Christian	0	0	2	13	2	17
Jewish	1	7	4	15	17	44
Muslim	0	0	0	2	1	3
Hindu	0	0	0	1	1	2
Two or More	1	1	0	0	1	3
Other	1	2	1	4	4	12
Total	7	19	25	104	69	224

Chi^2^ (32, 224) = 40.2, *p* = 0.152.

**Table 6 ijerph-20-07075-t006:** Religious affiliation and impact of religion on sexual health practices contingency table.

Religious Affiliation	Impact of Religion on Sexual Health Practices				
	Yes	No	Maybe	Unsure	Total
None	13	69	13	3	98
Catholic	10	16	9	2	37
Protestant	3	5	0	0	8
Other Christian	5	10	2	0	17
Jewish	1	32	8	3	44
Muslim	1	0	2	0	3
Hindu	0	1	1	0	2
Two or More	1	1	0	1	3
Other	4	5	3	0	12
Total	38	139	38	9	224

Chi^2^ (32, 224) = 48.2, *p* = 0.033.

**Table 7 ijerph-20-07075-t007:** Religiosity and current contraception method contingency table.

Time Spent Practicing Religion	Current Form of Contraception Method								
	None	Condoms	Pill	IUD	Implant	Plan B	>1	Other	Total
Never	9	13	17	17	6	0	44	1	107
Once or Twice/Year	4	10	10	13	3	0	20	1	61
At Least 5 Times/Year	2	4	3	3	1	0	16	0	29
Once a Month	1	0	1	1	0	1	7	0	11
Once a Week	1	0	0	3	0	0	5	0	9
Once a Day	0	0	1	2	1	0	3	0	7
Total	17	27	32	39	11	1	95	2	224

Chi^2^ (32, 224) = 37.9, *p* = 0.339.

**Table 8 ijerph-20-07075-t008:** Religiosity and age of first sex contingency table.

Religious Affiliation	Age of First Sex			
	Prior to 15	15 to 18	18 to 22	Total
Never	9	70	28	107
Once or Twice/Year	4	38	19	61
At Least 5 Times/Year	2	14	13	29
Once a Month	3	4	4	11
Once a Week	1	5	3	9
Once a Day	2	4	1	7
Total	21	135	68	224

Chi^2^ (10, 224) = 13.03, *p* = 0.222.

**Table 9 ijerph-20-07075-t009:** Religiosity and age of first contraception use contingency table.

Time Spent Practicing Religion	Age of First Contraception Usage				
	Never	Prior to 15	15 to 18	18 to 22	Total
Never	6	4	74	23	107
Once or Twice/Year	2	3	32	24	61
At Least 5 Times/Year	2	3	10	14	29
Once a Month	1	2	4	4	11
Once a Week	1	1	4	3	9
Once a Day	0	2	1	4	7
Total	12	15	125	72	224

Chi^2^ (15, 224) = 29.1, *p* = 0.016.

**Table 10 ijerph-20-07075-t010:** Religiosity and comfort discussing sexual health with parent/guardian contingency table.

Time Spent Practicing Religion	Comfort Discussing Sexual Health with Parent or Guardian					
	Extreme Discomfort	Mild Discomfort	Neither Discomfort nor Comfort	Mild Comfort	Extreme Comfort	Total
Never	20	35	9	28	15	107
Once or Twice/Year	9	22	4	18	8	61
At Least 5 Times/Year	4	11	5	5	4	29
Once a Month	2	3	2	2	2	11
Once a Week	1	4	0	3	1	9
Once a Day	4	0	1	2	0	7
Total	40	75	21	58	30	224

Chi^2^ (20, 224) = 17.3, *p* = 0.633.

**Table 11 ijerph-20-07075-t011:** Religiosity affiliation and comfort discussing sexual health with healthcare provider contingency table.

Time Spent Practicing Religion	Comfort Discussing Sexual Health with Healthcare Provider					
	Extreme Discomfort	Mild Discomfort	Neither Discomfort nor Comfort	Mild Comfort	Extreme Comfort	Total
Never	5	7	12	52	31	107
Once or Twice/Year	0	6	9	24	22	61
At Least 5 Times/Year	1	3	2	13	10	29
Once a Month	1	1	1	7	1	11
Once a Week	0	1	0	5	3	9
Once a Day	0	1	1	3	2	7
Total	7	19	25	104	69	224

Chi^2^ (20, 224) = 12.03, *p* = 0.915.

**Table 12 ijerph-20-07075-t012:** Religiosity and impact of religion on sexual health practices contingency table.

Religious Affiliation	Impact of Religion on Sexual Health Practices				
	Yes	No	Maybe	Unsure	Total
Never	17	70	17	3	107
Once or Twice/Year	6	44	8	3	61
At Least 5 Times/Year	6	14	7	2	29
Once a Month	2	5	3	1	11
Once a Week	3	5	1	0	9
Once a Day	4	1	2	0	7
Total	38	139	38	9	224

Chi^2^ (20, 224) = 40.06, *p* = 0.005.

## Data Availability

The data presented in this study are available upon request from the corresponding author.

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
