# Peer review of "An Analysis of the Impact of Religious Affiliation and Strength of Religiosity on Sexual Health Practices of Sexually Active Female College Students"

_ijerph, 2023, doi:10.3390/ijerph20227075_

Round 1

Reviewer 1 Report (Previous Reviewer 1)

Comments and Suggestions for Authors

I appreciate how you responded to my concerns. Your edits and additions clarify the focus and findings.

Author Response

We thank the reviewer for their efforts in improving our manuscript.

Reviewer 2 Report (New Reviewer)

Comments and Suggestions for Authors -It would be helpful to understand the rationale for how the authors defined sexual activity in this study and how that relates to the literature on what defines sexual activity.  -Why the researchers chose the angle of religiosity given the lack of students practicing in a religion regularly is not well described and/or understood in the context of this study. It seems a different possible determinant or factors impacting sexual health would have led to more impact in this study, so why religiosity as the most important factor to study should be better discussed. -The article discusses teen pregnancy and research around adolescent sexual health; however given this research was done with college going females these associated life stages may not apply as well given the nature of the college environment and the age ranges used for this study that fall outside of these life stage definitions. It would be helpful for the authors to discuss this in the context of the discussion and/or limitations, strengths and broader PH implications. 

Author Response

GENERAL RESPONSE: Thank you for the constructive feedback of the reviewer and the collegial dialogue.

COMMENT: It would be helpful to understand the rationale for how the authors defined sexual activity in this study and how that relates to the literature on what defines sexual activity.

RESPONSE: While the topic of adolescent sexual activity has been a popular area of focus, an explicit definition of sexual activity is rarely seen in previous literature. Seldomly, a general definition from a dictionary or encyclopedia is used as a model to build a definition that fits with the parameters of the previous literature’s study design(1,5,8). Traditionally, the definition of sexual activity is limited to the heterosexual activity of penile penetration of a vagina (Britannica ). However, as the 21st century has progressed, a more functional definition of sexual activity has developed to include sexual acts outside of heterosexual penetration. According to the Britannica Encyclopedia, sexual activity is defined as “any activity—solitary, between two persons, or in a group—that induces sexual arousal” (Britannica). Based on this general description, the present study built the “sexual activity” variable based on two key parameters: the type of action and the duration of the activity. The present study aimed to analyze how religiosity impacted the use of contraceptive methods as a means to prevent pregnancy and/or STI transmission. Therefore, while previous literature has determined that sexual activity can include any act that promotes sexual arousal, the focus was kept on oral sex and vaginal or anal penetration because those activities pose a large threat to STI transmission and pregnancy. In addition, the definition of sexual activity as seen in previous literature does not often specify a duration, however for the present study, participants were included only if they had participated in sexual activity at least 3 times in the last 3 months to assure that they had consistency in their sexual health behaviors. We have revised the methods to reflect our intent better.

Citation:

Human sexual activity [Online]. Encyclopædia Britannica Encyclopædia Britannica, inc.: [date unknown]. https://www.britannica.com/topic/human-sexual-activity/Sociosexual-activity

COMMENT: Why the researchers chose the angle of religiosity given the lack of students practicing in a religion regularly is not well described and/or understood in the context of this study. It seems a different possible determinant or factors impacting sexual health would have led to more impact in this study, so why religiosity is the most important factor to study should be better discussed. 

RESPONSE: It is important to note that religious upbringing and the values instilled during childhood could still play a role despite the lack of current practice, as we did find a significant impact of religion. Further, even though Skidmore College is not religiously affiliated and very few participants currently practice a religion, religion is a part of the student life at Skidmore. There is a wide variety of religious clubs on campus that have hundreds of members and hold well-attended events; the lack of religiosity seen in the present study was greater than expected based on the success of religious clubs on campus. In addition, religiosity was determined to be an important factor to study because it was a variable that had never been highlighted before on campus, making this study design highly warranted. As previously mentioned, Skidmore College has many different programs that focus on sexual health, such as seminars on consent, sexual assault, and STI transmission, from a medical or scientific standpoint. Seldom was sexual health talked about on campus in the context of religiosity.  We have revised the manuscript strategically to reflect this intent.

COMMENT: The article discusses teen pregnancy and research around adolescent sexual health; however, given this research was done with college-going females these associated life stages may not apply as well given the nature of the college environment and the age ranges used for this study that fall outside of these life stage definitions

RESPONSE: While we aren’t sure exactly the exact intent of the reviewer, the sample did factually include women in their teens (18-19) and those of childbearing age. The present study sampled college aged females as a means to study adolescent sexual health behaviors. As defined by the World Health Organization, an adolescent is a person between childhood and adulthood, often quantified as between the ages of 15-19, and adolescent pregnancy is defined as pregnancy that occurs in a woman younger than 20 (WHO #). However, previous literature has demonstrated that this definition is fluid, and the exact age range can vary based on author's interpretation. Therefore, the age range of participants included in the present study was 18-22 years old, which does include the higher end of the WHO definition spectrum and therefore does allow the results to be generalized for the adolescent or teen pregnancy population. However, the present study does include participants who may be older than some may consider an adolescent, therefore subsequent research that focuses on a younger partisan pool, such as high school students, is warranted to study the lower end of the adolescent age range.

Citation:

Adolescent pregnancy [Online]. World Health Organization World Health Organization: [date unknown]. https://www.who.int/news-room/fact-sheets/detail/adolescent-pregnancy [30 Oct. 2023].

This manuscript is a resubmission of an earlier submission. The following is a list of the peer review reports and author responses from that submission.

Round 1

Reviewer 1 Report

Comments and Suggestions for Authors

I appreciated reading your manuscript. Thanks for your work in this area. While I thought that much of what you did made good sense given your research question, I found one significant flaw in your method. The fact that you did not include those who were abstinent meant that you excluded a large group of Christian students. Abstinence is conceptualized as a method of contraception for many. I encourage you to include that group of students in your analyses.

More minor recommendations include:

1) I would use the term "college students" rather than "adolescents." In the U.S., "adolescents" typically refer to those who are in secondary school.

2) I see a need to elaborate more on comments you made about how contraception can be viewed as a sin by some, how some religions enforce moral behavior via heaven/hell, etc. 

3) I don't understand how your study is about female students (as per the title) yet your sample included some who identified in another way (Figure 3).

Author Response

Reviewer 1

I appreciated reading your manuscript. Thanks for your work in this area. While I thought that much of what you did made good sense given your research question, I found one significant flaw in your method. The fact that you did not include those who were abstinent meant that you excluded a large group of Christian students. Abstinence is conceptualized as a method of contraception for many. I encourage you to include that group of students in your analyses.

REPLY: Thank you for your support and constructive review of the present manuscript. We have added a statement regarding including those abstinent in future studies as well as studying colleges/universities that are religiously affiliated to increase representation to better explore such relations. However, nearly 20% of the sample identified as Christian (or as one of their religions), and the sample does represent likely well the women at a liberal arts institution. Thus, while perhaps not addressing college-age students in its entirety, we hope the reviewer can agree that the study does help to answer the research question, at least in the current context.

More minor recommendations include:

1) I would use the term "college students" rather than "adolescents." In the U.S., "adolescents" typically refer to those who are in secondary school.

REPLY: Thank you for this suggestion, we have changed terms, where appropriate, unless referring to prior literature where adolescents is appropriate.

2) I see a need to elaborate more on comments you made about how contraception can be viewed as a sin by some, how some religions enforce moral behavior via heaven/hell, etc. 

REPLY: Thank you for this insightful comment, we have edited the manuscript for clarity on this issue in the introduction and the discussion.

3) I don't understand how your study is about female students (as per the title) yet your sample included some who identified in another way (Figure 3).

REPLY: We apologize for any confusion, the only exclusions that were made were excluding those who identified as biologically male. Accordingly, 88% of respondents were cis-gender female, otherwise, reported as students who were determined as female biological sex at birth, but may identify as other gendered, this is now highlighted in the methods. 

Reviewer 2 Report

Comments and Suggestions for Authors

I think this is an important article.  It however is conducted at a small, liberal arts college with perceived progressive stance.  The students are not very active in religion, yet the results are suggestive.  I would love to see a cross college study of 2 or 3 religiously affiliated schools with schools like Skidmore and perhaps a couple public universities.  The findings are limited, yet interesting.  They do however set a model and trajectory for future research.  It that future research is conducted then publishing the study will be of value. FYI the research design and care is excellent.

Author Response

Reviewer 2

I think this is an important article.  It however is conducted at a small, liberal arts college with perceived progressive stance.  The students are not very active in religion, yet the results are suggestive.  I would love to see a cross college study of 2 or 3 religiously affiliated schools with schools like Skidmore and perhaps a couple public universities.  The findings are limited, yet interesting.  They do however set a model and trajectory for future research.  It that future research is conducted then publishing the study will be of value. FYI the research design and care is excellent.

REPLY: We would like to thank the reviewer for their support and constructive review of the present manuscript. As suggested by the reviewer, we have added a statement regarding future studies at colleges or universities that are religiously affiliated to increase representation to better explore such relations. Though it worth noting that nearly 20% of the sample identified as Christian (or as one of their religions), and the sample does represent likely well the women at a liberal arts institution, thus while perhaps not addressing college students in its entirety, or of those at a religiously affiliated higher education institution.  the study does help to answer the research question, at least in the current context.

Round 2

Reviewer 1 Report

Comments and Suggestions for Authors

Thanks for your adjustments to your manuscript. However, my initial concern about still stands: the lack of representation of those who abstain as a form of contraception (largely evangelical Christians). 

Author Response

Please see the reply in the attachment.
